# Perfringolysin O-Induced Plasma Membrane Pores Trigger Actomyosin Remodeling and Endoplasmic Reticulum Redistribution

**DOI:** 10.3390/toxins11070419

**Published:** 2019-07-17

**Authors:** Cláudia Brito, Francisco S. Mesquita, Christopher K. E. Bleck, James R. Sellers, Didier Cabanes, Sandra Sousa

**Affiliations:** 1i3S-Instituto de Investigação e Inovação em Saúde, IBMC, Universidade do Porto, 4099-002 Porto, Portugal; 2Programa Doutoral em Biologia Molecular e Celular (MCbiology), Instituto de Ciências Biomédicas Abel, Salazar, Universidade do Porto, 4099-002 Porto, Portugal; 3Electron Microscopy Core, National Heart, Lung, and Blood Institute, National Institutes of Health, Bethesda, MD 20892, USA; 4Laboratory of Molecular Physiology, National Heart, Lung, and Blood Institute, National Institutes of Health, Bethesda, MD 20892, USA

**Keywords:** pore-forming toxins, perfringolysin O, actomyosin remodeling, plasma membrane damage

## Abstract

*Clostridium perfringens* produces an arsenal of toxins that act together to cause severe infections in humans and livestock animals. Perfringolysin O (PFO) is a cholesterol-dependent pore-forming toxin encoded in the chromosome of virtually all *C. perfringens* strains and acts in synergy with other toxins to determine the outcome of the infection. However, its individual contribution to the disease is poorly understood. Here, we intoxicated human epithelial and endothelial cells with purified PFO to evaluate the host cytoskeletal responses to PFO-induced damage. We found that, at sub-lytic concentrations, PFO induces a profound reorganization of the actomyosin cytoskeleton culminating into the assembly of well-defined cortical actomyosin structures at sites of plasma membrane (PM) remodeling. The assembly of such structures occurs concomitantly with the loss of the PM integrity and requires pore-formation, calcium influx, and myosin II activity. The recovery from the PM damage occurs simultaneously with the disassembly of cortical structures. PFO also targets the endoplasmic reticulum (ER) by inducing its disruption and vacuolation. ER-enriched vacuoles were detected at the cell cortex within the PFO-induced actomyosin structures. These cellular events suggest the targeting of the endothelium integrity at early stages of *C. perfringens* infection, in which secreted PFO is at sub-lytic concentrations.

## 1. Introduction

*Clostridium perfringens* is an anaerobic gram-positive bacterium often found as a natural resident of the intestine of humans and animals [1,2]. Under poorly characterized stimuli, this bacterium undergoes rapid proliferation and causes severe systemic and enteric diseases such as gas gangrene (myonecrosis), gastroenteritis, and enterocolitis [3,4]. The pathogenicity of *C. perfringens* is largely attributed to its ability to produce an arsenal of toxins [5]. While many of its toxins are encoded in extra-chromosomal plasmids, the gene encoding the pore-forming toxin (PFT), perfringolysin O (PFO) [6], is located in the chromosome of the majority of *C. perfringens* strains [7]. PFO has been extensively studied and became the structural archetype of PFTs belonging to the cholesterol-dependent cytolysin (CDCs) family [8,9], which comprises a variety of PFTs such as listeriolysin O (LLO), produced by *Listeria monocytogenes* or pneumolysin (PLY), produced by *Streptococcus pneumoniae.* PFO is secreted by *C. perfringens* as a water-soluble monomer that upon binding to the host plasma membrane (PM) oligomerize and assemble into transmembrane pores [10,11]. Such pores allow the uncontrolled flux of ions and small molecules between extracellular and intracellular milieus, disrupting the cell homeostasis and triggering host defense mechanisms to repair the inflicted damage and recover cell homeostasis [12]. Although the mechanism of action of PFO and several other PFTs have been extensively studied (reviewed in [13]), less is known about the protective cellular responses engaged to counteract intoxication.

Despite the recognized importance of PFO in *C. perfringens* pathogenesis, its individual and specific role in disease remains poorly understood [11]. PFO has shown to act in synergy with the clostridial α-toxin in gas gangrene and hemorrhagic enteritis in calves [14,15,16]. PFO also induces TLR4-dependent macrophage activation followed by the expression of pro-inflammatory cytokines [17] and the expression of adhesion molecules in both endothelial cells [18] and polymorphonuclear leukocytes [19]. These events lead to the aggregation of platelets and leukocytes that ultimately obstruct the blood vessels and impair the migration of inflammatory cells to the sites of infection, which are the hallmarks of gas gangrene [18,20,21]. Furthermore, PFO together with α-toxin have a direct cytotoxic effect on endothelial cells and leukocytes [22,23]. PFO thus appears to potentiate the role of other clostridial toxins allowing the bacteria to rapidly grow in tissues and escape from host response mechanisms, causing extensive necrosis and systemic toxemia [11]. Despite such findings, more studies are needed to fully understand the role of PFO during *C. perfringens* infections. In particular, analyzing the consequences of PFO-induced pore-formation at the cellular level, at low PFO concentrations, can give important insights on its role at early stages of the infection, when the toxin acts locally and without reaching the systemic circulation.

Here we analyze at the cellular level, the host responses to PFO attack. Given that PFTs induce actomyosin rearrangements and endoplasmic reticulum (ER) vacuolation [12,24,25], which may affect the ability of the injured cells to migrate and alter the surface exposed proteins, we assessed the cytoskeletal responses and ER alterations in response to PFO.

We show that PFO intoxication lead to the remodeling of the cytoskeleton, inducing a calcium-dependent assembly of compact and well-defined actomyosin structures at the cell cortex. We further demonstrate that the assembly of these actomyosin structures, previously observed in response to LLO intoxication [24], occurs with specific features in human endothelial cells, and correlates with the host cell recovery from PFO-induced PM damage. Moreover, we show that PFO not only triggers the rearrangement of the actomyosin cytoskeleton but also induces ER disruption. Our work demonstrates that PFO attack triggers cell autonomous cytoskeletal rearrangements that may contribute to recover homeostasis.

## 2. Results

### 2.1. PFO Induces the Assembly of Actomyosin-Enriched Structures at the Cell Cortex

To evaluate the effects of purified PFO on cultured epithelial cell monolayers, we treated HeLa cells with growing concentrations of PFO. Intoxicated cells showed massive morphological alterations detected by phase-contrast microscopy (Figure 1A). As compared to non-treated conditions, dramatic PM blebbing was observed in PFO-intoxicated cells (Figure 1A), suggesting the underlying remodeling of the actomyosin cytoskeleton to support such PM blebbing. To assess the cytoskeletal remodeling triggered by PFO, we analyzed the redistribution of actin and the heavy chain of myosin IIA (NMHCIIA) by confocal immunofluorescence microscopy. PFO-intoxicated cells showed strong reorganization of the actomyosin network (Figure 1B and Appendix A). We detected well-defined cortical structures arranged in bundles highly enriched in actin and myosin IIA (Figure 1B). Such structures are reminiscent of actomyosin bundles reported in LLO-intoxicated cells [24], which assemble at sites of the PFT-induced PM blebbing. The percentage of cells showing cortical actomyosin bundles increased with growing concentrations of PFO (Figure 1C), suggesting that the assembly of these structures is correlated with increased PFO-induced cell damage. We followed the assembly of PFO-induced actomyosin structures by live-cell imaging of HeLa cells co-expressing NMHCIIA fused to green fluorescent protein (GFP) and tdTomato-F-Tractin (Figure 1D, Appendix A). Shortly after PFO addition, cells contracted and the cortical actomyosin cytoskeleton disassembled. These events were followed by transient perinuclear actomyosin polymerization, intense membrane associated ruffling, lamellipodia formation, and reassembly of actomyosin cytoskeleton in well-defined cortical structures (Figure 1B,D and Appendix A).

Given that during pathogenesis PFO would target preferentially the endothelial cells to promote the development of gas gangrene [18], we tested whether the assembly of actomyosin structures identified in HeLa cells also occur in a more pathophysiological relevant system. Human umbilical vein endothelial cells (HUVECs) were incubated with growing concentrations of PFO for 20 min, fixed and processed for confocal immunofluorescence microscopy to detect actin and NMHCIIA. Compact cortical structures enriched in myosin IIA were detected in PFO-intoxicated HUVECs (Figure 2A). Here, the structures appeared organized in clusters of myosin rings resembling honeycombs also described at actomyosin dependent secretion sites during endothelial exocytosis [26,27]. Accumulation of F-actin in such structures was less evident (Figure 2A). As determined in the HeLa cells, the percentage of HUVECs showing cortical myosin IIA bundles increased with growing concentrations of PFO (Figure 2B), again suggesting that the assembly of NMHCIIA-enriched structures correlates with PFO-induced cell damage. In addition, the assembly of actomyosin cortical structures occurs in different cell lines supporting a broad role for this event.

### 2.2. Loss of PM Integrity is Concomitant with the Assembly of Cortical Actomyosin Structures

We further assessed whether the assembly of actomyosin cortical bundles correlates with PFO-induced cell damage. HeLa cells were intoxicated with increasing concentrations of PFO ranging from 0.01 to 0.5 nM and PM integrity was assessed by propidium iodide (PI) permeability assays under flow cytometry analysis. The percentage of PI positive cells increased in response to the growing concentrations of PFO, indicating that the disruption of the PM integrity is PFO dose-dependent (Figure 3). Thus, the percentage of cells permeable to PI and displaying cortical actomyosin bundles increased in response to increasing concentrations of PFO. In particular, under 0.5 nM PFO approximately 40% of the cells were permeable to PI and simultaneously, actomyosin bundles were detected in ≈ 40% of the cells (Figure 1C and Figure 3). Together, these data suggest that actomyosin bundles assemble at the cortex of the cells concomitantly to the loss of PM integrity triggered by PFO pore formation. In addition, our results point to the involvement of actomyosin remodeling in PM repair mechanisms induced by PFO, as shown previously for other PFTs [12,24].

### 2.3. Assembly of Actomyosin Bundles Requires Pore Formation, Calcium Influx and Myosin II Activity

As previously described for LLO [24], our observations in PFO-intoxicated cells led us to hypothesized that the assembly of PFO-induced actomyosin bundles depends on pore formation and associated secondary events, such as the raise in intracellular calcium concentration. To test this, HeLa cells were intoxicated with PFO in different conditions and the percentage of cells showing actomyosin structures and permeability to PI were determined. Cell intoxication with PFO pre-incubated with cholesterol, which blocks pore formation [28], strongly impaired the assembly of actomyosin structures (Figure 4A,B). In agreement with a defective pore-forming activity, the percentage of PI permeable cells was reduced as compared to control conditions in which HeLa cells were intoxicated with PFO diluted in HBSS (Figure 4C). These data confirm that the assembly of actomyosin cortical structures is dependent on pore formation.

To evaluate whether the assembly of actomyosin cortical structures was dependent on the influx of calcium, we incubated HeLa cells with PFO in a calcium-free medium. In such conditions, actomyosin cortical structures failed to assemble (Figure 4A,B). In addition, we found that the assembly of actomyosin structures is reduced in the presence of blebbistatin, a pharmacological inhibitor of myosin II activity [29]. Thus, the actomyosin remodeling induced by PFO at the cortex of intoxicated cells requires both the influx of calcium and myosin II activity. The percentage of PI-positive cells was not statistically different in control conditions and in populations intoxicated under calcium free conditions or blebbistatin treatment (Figure 4D), indicating that calcium influx and myosin II activity are both dispensable for pore formation. However, at the time point of analysis, we noticed a consistent slight increase in both the percentage of PI-positive and the mean fluorescence intensity of cells intoxicated in calcium free conditions or in the presence of blebbistatin. These observations are in agreement with the well-documented role of calcium and myosin IIA in repair responses to PM damage [12,24]. Possibly, the increase in PI-positive cells will be more evident after toxin washout or at later time points, since control cells repair PM wounds whereas under calcium free or blebbistatin conditions cells accumulate PM damage and fail to recover. Together, these data suggest a role for actomyosin structures in the cellular response triggered by the PFO-induced pores.

### 2.4. Recovery of PM Integrity Occurs Simultaneously with the Resolution of Actomyosin Structures

We further explored the role of actomyosin structures in the cellular response to PFO-induced PM damage. For that, we intoxicated HeLa cells with 0.1 nM PFO for 10 min and washed out the unbound toxin to allow cells to recover. At different recovery time points ranging from 0 to 24 h, cells were fixed and processed for immunofluorescence analysis and the percentage of cells displaying actomyosin structures was quantified. Consistently with data in Figure 1C, after PFO intoxication about 30% of the cells exhibited cortical actomyosin rearrangements (Figure 5A,B). The percentage of cells displaying actomyosin bundles decreased progressively throughout the time of recovery and reached 10% of the total population after 24 h (Figure 5A,B), indicating that actomyosin structures disassemble following PFO washout. To confirm that the decreased percentage of cells displaying actomyosin bundles was indeed related to the progressive disassembly of structures and not because of cell death, we assessed the total number of cells along time after washout. The number of cells per field was quantified under the microscope in the different conditions: non-intoxicated (NT), intoxicated without recovery (0 h), and after washout (from 1 to 24 h). No differences were detected (Figure 5C), indicating that the cells were not detaching throughout the time. Together, these results indicate that the assembly of actomyosin structures is a reversible process and suggest that such bundles disassemble as the PM recovers its integrity.

To assess the correlation between the disassembly of actomyosin structures and the reseal of the PM pores and the consequent recovery of PM integrity, we determined the percentage of PI-positive cells after PFO washout. In agreement with data in Figure 3, 30% of the cells were permeable to PI after intoxication with 0.1 nM PFO (Figure 5D). The percentage of PI-positive cells also decreased progressively throughout the time of recovery reaching the levels of non-intoxicated cells (NT) between 12 to 24 h after washout (Figure 5D). Our data thus demonstrate that the disassembly of actomyosin structures and the restoration of PM integrity follow similar kinetics, indicating that PFO-induced cortical actomyosin rearrangements contribute to the restoration of PM integrity, as described for the other PFTs [24].

### 2.5. PFO Induces ER Redistribution and Accumulation within PM Enclosed Actomyosin Structures

We aimed to gain further knowledge on the nature of the actomyosin response to PFO attack. We assessed whether PFO-induced actomyosin rearrangements were similar to the assembly of cortical actomyosin structures previously described in response to other PFTs [24]. PM labeling with fluorescently conjugated wheat germ agglutinin (WGA) showed that actomyosin structures assemble just beneath the PM in PFO-intoxicated cells (Figure 6A), presumably to assist PM remodeling required to face damage and achieve repair.

Given that PFTs were described to target the endoplasmic reticulum (ER) [24,30,31], we evaluated the structure and sub-cellular localization of the ER in PFO-intoxicated cells as compared to control non-treated cells. Cells were processed for immunofluorescence and immunolabeled to detect NMHCIIA and ER proteins, using an anti-KDEL antibody, which detects the Lys-Asp-Glu-Leu (KDEL) sequence present at the carboxy-terminus of various ER resident proteins. Control non-treated cells showed a characteristic ER distribution, with a perinuclear concentration and a fine reticular network that spans to the cell periphery. In contrast, PFO-intoxicated cells displayed vacuolation of the ER with vacuoles expanding across the cytoplasm to the cortex (Figure 6B). ER-positive vacuoles accumulated at both cortical actomyosin structures and around the perinuclear region where they were tightly surrounded by myosin IIA (Figure 6B). Transmission electron microscopy (TEM) analysis of HeLa cells ectopically expressing an ER-retained KDEL-GFP fusion protein confirmed the presence of intracellular vacuoles enriched in ER components in intoxicated cells as detected by GFP immunogold labeling (Figure 6C). Such ER vacuoles were absent in control non-treated cells, in which KDEL-GFP immunodetection by TEM revealed homogeneous distribution of the ER within the cytoplasm (Figure 6C).

Gp96, an ER chaperone [32], was previously described to interact with NMHCIIA in response to PFT attack and to be involved in the assembly and dynamics of PFTs-induced actomyosin cortical structures [24]. Considering that the ER accumulates at the PFO-induced actomyosin bundles, we evaluated the role of Gp96 and NMHCIIA in the assembly of such structures. HeLa cells expressing control oligonucleotides (shControl) or targeting the expression of Gp96 or NMHCIIA (shGp96 and shNMHCIIA) were intoxicated with PFO (0.1 nM for 20 min), fixed and processed for immunofluorescence detection of actin and NMHCIIA (Figure 6D). The percentage of cells displaying PFO-induced actomyosin structures was decreased for shGp96 and shNMHCIIA cells (Figure 6E), indicating that the expression of both Gp96 and NMHCIIA is required for the assembly of actomyosin structures induced by PFO intoxication. Together with published data, our results demonstrate the involvement of the actomyosin cytoskeleton and ER components in the cellular response to the attack by several PFTs, and support the existence of a common cell response mechanism to face damage induced by CDCs.

## 3. Discussion

Following PFTs-induced damage, host cells employ repair processes and exhibit profound cytoskeletal remodeling and simultaneous PM blebbing. Although the control of the cytoskeletal organization is necessary to promote PM recovery [33], the role of the actomyosin cytoskeleton dynamics in PFT-intoxicated cells remains poorly studied. Here, we investigate the host cytoskeletal responses to PFO attack. We found that, at sub-lytic concentrations, PFO induces a profound reorganization of the actomyosin cytoskeleton culminating in the assembly of cortical actomyosin structures at sites of PM blebbing in both HeLa and HUVEC cells. Blebbing occurs during several physiological processes such as cell migration and cytokinesis and it has also been described as a common cellular response to PM damage [34,35]. PM blebs are expanded PM protrusions resulting from the intracellular pressure created by the detachment of the PM from the cytoskeletal cortex [34,35]. They eventually retract through a mechanism that is dependent on NMII activity and actomyosin contraction or can be extruded in response to PFTs [25,31,36]. PFT-induced PM blebbing is thought to be protective by quarantining injured sites, preventing excess calcium influx and loss of cytosolic content [36,37], and eventually by removing the PFT pores if the blebs are released. Alternatively, PM blebbing may act as a secondary event of the PFT-induced cortical cytoskeletal remodeling. During the expansion and retraction of PM blebs, the actomyosin cytoskeleton is very dynamic, undergoes significant reorganization, and assembles into bundles that are structurally similar to the PFO-induced actomyosin structures characterized here [38].

The remodeling of the actomyosin cytoskeleton triggered by PFTs occurs mainly through the sustained rise in cytosolic calcium levels which follows PM damage. A similar mechanism was described during cell adaptation responses to a variety of stimuli that induce cytosolic calcium rise. This process relies on the activation of the ER-associated inverted formin 2 (IFN2), which triggers the ER and actin cytoskeleton remodeling. This reorganization facilitates cell adaptation and recovery responses to different types of PM damage, wound healing, and cell migration [39]. Accordingly, our data also shows that the actomyosin remodeling induced by PFO at the cortex of intoxicated cells requires both pore-formation and the influx of calcium. It is likely that the uncontrolled ion flux upon pore formation alters the intracellular ionic strength, which critically interferes with the dynamics of NMHCIIA filament in vitro [40]. In addition, ATP that was shown to diffuse out of the cells upon PFT intoxication [12], also modulates NMHCIIA dynamics in vitro [40]. Whether possible alterations in ionic strength and ATP diffusion will molecularly control the assembly of actomyosin bundles upon PFT attack remains to be evaluated.

Following PFT intoxication of HeLa cells in vitro, the actomyosin network recovers the normal organization within ~4 to 8 h, with the occasional release of large cytoplasmic containing blebs [24,25]. Upon PFT intoxication in vivo, gut epithelial cells contract reducing the thickness of epithelial layer while maintaining barrier function. This process also involves massive actomyosin remodeling, purging of damaged organelles, and subsequent recovery of normal cell morphology [31]. Similarly, we demonstrate that the assembly of actomyosin structures triggered by PFO intoxication is a transient and reversible process. The disassemble of such structures follows the recovery of the PM integrity, which occurs within hours after PFO washout.

Interestingly, we also demonstrate the interconnection between the actomyosin cytoskeleton and ER components in response to PFO attack. We showed that PFO induces the disruption of the ER and the accumulation of ER-vacuoles within cortical actomyosin structures. The ER chaperone Gp96 controls the assembly of the PFO-induced actomyosin bundles suggesting that it regulates cytoskeletal remodeling in response to the PFO-induced damage, as previously shown during LLO-dependent *L. monocytogenes* infection. Gp96 interacts with myosin IIA and the actin adaptor Filamin A, regulates cytoskeletal remodeling in response to LLO pore-formation and is critical for host survival upon LLO-induced damage [24]. Together these studies demonstrate that Gp96 and myosin IIA interaction coordinates cytoskeletal dynamics necessary for efficient PM repair in cells intoxicated with several different PFTs. Furthermore, Gp96 has been involved in processes remarkably dependent on cytoskeletal organization and dynamics such as cell migration and cell polarity [41], emphasizing the role of this ER chaperone as a cytoskeletal regulator.

PFO-induced actomyosin bundling was observed in both epithelial and endothelial cells, indicating that such cellular events may affect both epithelial and endothelial organization. However, how these events define *C. perfringens* infection remains unknown. It is plausible that during the early stages of infection low concentrations of PFO induce local cytoskeleton alterations and release cytosolic material, which enhances endothelium dysfunction and potentiates the action of other clostridial toxins. Together these events will promote inflammation, reduce vascular resistance, and contribute to the vasodilatation during necrohemorrhagic enteritis [11], ultimately determining the onset of the disease.

## 4. Material and Methods

### 4.1. Plasmids, Antibodies, and Dyes

Plasmid allowing the expression of PFO in *E. coli* was kindly offered by Daniel Portnoy (DP-4167) [42]. Plasmid GFPNMHCIIA was a gift from Robert Adelstein through Addgene (#11347) [43], plasmid tdTomato-F-Tractin [44] was kindly offered by John Hammer (NIH, Bethesda, MD, USA), and plasmid pUBC eGFP-KDEL was a gift from François-Xavier Campbell-Valois [45]. The following antibodies were used at 1/200 for immunofluorescence microscopy (IF): rabbit anti-NMHCIIA (Sigma, St Louis, MO, USA), mouse anti-NMHCIIA (Abcam, Cambridge, UK), and mouse anti-KDEL (Abcam, Cambridge, UK). Secondary antibodies were used at 1/200: goat anti-rabbit Alexa Fluor 488 (Invitrogen, Waltham, MA, USA) and goat anti-mouse Cy3 (Jackson ImmunoResearch, West Grove, PA, USA). For IF, F-actin was labeled with rhodamine phalloidin (Invitrogen, Waltham, MA, USA) and the PM with FITC-conjugated WGA (Sigma, St Louis, MO, USA) that recognizes sialic acid and N-acetylglucosaminyl sugar residues at the PM.

### 4.2. Cell Lines and Reagents

HeLa cells (ATCC CCL-2) (ATCC, Manassas, VA, USA) were grown in DMEM with glucose and L-glutamine, supplemented with 10% fetal bovine serum (FBS; Biowest, Nuaillé, France). HUVEC cells (ATCC CRL-1730) were grown in M199 supplemented with 10% FBS, 100 µg/mL Heparin, and 60 µg/mL of the endothelial cell growth factor, added fresh during cell seeding. HUVEC cells were seeded on flasks previously coated with 0.2% gelatin (2 mL for 30 min). HeLa cells depleted for the expression of Gp96 or NMHCIIA were previously generated [24]. The lentiviral shRNA expression plasmids Mission pLKO.1-puro (control) and Mission shRNA-Gp96 or shRNA-Myh9 (Sigma, St Louis, MO, USA) were used in combination with the envelope plasmid pMD.G and packaging plasmid pCMVR8.91 and co-transfected into HEK293 cells. Viral supernatants were harvested after 72 h, filtered and incubated with target HeLa cells for 48 h. Puromycin was used for selection of cell lines. Cell culture media and supplements were from Lonza (Basel, Switzerland) and Sigma (St Louis, MO, USA). Cells were maintained at 37 °C in a 5% CO_2_ atmosphere. Drug and toxin treatments and washes were carried out in Hanks’ balanced salt solution (HBSS) or, when indicated, in HBSS Ca^2+^-free medium. Blebbistatin (Sigma, St Louis, MO, USA) was used at 25 μM for 30 min prior to PFO intoxication and maintained throughout the treatment. Cholesterol (Sigma, St Louis, MO, USA) was used as described previously [28].

### 4.3. Toxins and Intoxications

PFO was purified as previously described [42]. Briefly, PFO expression was induced in *E. coli* BL21 (DE3 PlysS) with 1mM IPTG, at 30 °C for 6 h. PFO-expressing bacteria were harvested by centrifugation, resuspended in cold lysis buffer (50 mM sodium phosphate, pH 8.0, 1 M sodium chloride, 20 mM imidazole, 10 mM 2-mercaptoethanol, 1 mM AEBSF), and lysed in a French pressure cell. Bacterial lysate was centrifuged (17,000× *g*, 40 min, 4 °C) and the supernatant was incubated (1 h, 4 °C) with Ni-NTA resin (QIAGEN, Venlo, The Netherlands). The resin was washed with wash buffer (lysis buffer, pH 6.0, 10% glycerol, 0.1% Tween 20) and proteins were eluted in elution buffer (lysis buffer, pH 6.0, and 400 mM imidazole). Eluted fractions were dialyzed overnight in storage buffer (lysis buffer, pH 6.0, with 1 mM EDTA) at 4 °C. Aliquots of purified and concentrated PFO were directly stored at −80 °C.

Unless mentioned, intoxications were performed in 24-well plates with 1 ml at the indicated toxin concentrations and durations. HeLa cells and HUVECs were seeded at 5 × 10^4^ per well. For recovery assays, cells were washed three times in HBSS after PFO intoxication and allowed to recover in culture medium supplemented with 10% of FBS for the indicated times.

### 4.4. Immunofluorescence Microscopy

Cells were fixed in 4% paraformaldehyde (20 min), quenched with 0.1 M NH_4_Cl (40 min), stained with FITC-conjugated WGA for 1 h (whenever indicated), and permeabilized with 0.1% Triton X-100 (5 min). Antibodies were diluted in 1x PBS containing 1% BSA. Coverslips were incubated for 1 h with primary antibodies, washed three times in PBS and incubated 45 min with secondary antibodies and rhodamine phalloidin. Coverslips were mounted onto microscope slides with Aqua-Poly/Mount (Polysciences, Warrington, PA, USA). Images were collected with an epifluorescent Olympus BX53 microscope equipped with a 20 × 0.75 NA objective or a Leica SP5II confocal laser-scanning microscope equipped with a 63 × 1.4 NA objective and processed using ImageJ [46] and Adobe Photoshop software.

### 4.5. Immunofluorescence Quantifications

Unless indicated, IF quantifications were performed in at least 200 cells per sample and in at least three independent experiments. For actomyosin bundles, cells were scored positive when displaying at least one compact myosin IIA bundle. DAPI nuclear staining was used for the quantification of the number of cells per field. Ten fields per condition were quantified in three independent experiments.

### 4.6. Transmission Electron Microscopy Analysis

Non-treated and 0.2 nM PFO samples were fixed in 4% formaldehyde, 0.1% glutaraldehyde (GA) in 60 mM PIPES, 25 mM HEPES, 10 mM EGTA, and 1 mM MgCl_2_ (pH 6.9) (PHEM buffer, pH 6.9) [47] for 90 min. Cells were scraped and sedimented at 250*g*. Cryosectioning and immunolabeling were performed as described elsewhere [48,49]. In brief, ultrathin sections (50–60 nm) from gelatin-embedded and frozen cell pellets were obtained using an FC7/UC7-ultramicrotome (Leica, Wetzlar, Germany). Immunogold labeling was carried out on thawed sections with an anti-GFP antibody (1:200, Rockland Immunochemicals, Limerick, PA, USA) and 10 nm protein A-gold (1:50, UMC Utrecht University, Utrecht, The Netherlands) as described [48,50] and stained/embedded in 4% uranyl acetate/2% methylcellulose mixture (ratio 1:9) [51]. Thin sections were examined on a JEM-1200EX (JEOL, Tokyo, Japan) transmission electron microscope (accelerating voltage 80 keV) equipped with an AMT 6-megapixel digital camera (Advanced Microscopy Techniques Corp, Woburn, MA, USA).

### 4.7. Live Imaging PFO-Treated Cells

HeLa cells seeded (1 × 10^5^) into Lab Tek 1.5 glass chamber (4 μ-wells) were transfected to simultaneously express GFPNMHCIIA and tdTomato-F-Tractin and maintained in HBSS at 37 °C with 5% CO_2_. Transfections were performed with 0.5 µg of each plasmid DNA using lipofectamine 2000 (Thermo Fisher Scientific, Waltham, MA, USA) according to manufacturer’s instructions. Cells were imaged using a super-resolution mode on a Zeiss LSM 880 Airyscan microscope equipped with a 63 × 1.4 NA objective. PFO (0.2 nM) was added right after initial image acquisition. Raw data was processed using Airyscan processing. ImageJ [46] was used for image sequence analysis, video assembly, and brightness and contrast adjustments.

### 4.8. Flow Cytometry Analysis of PFO-Treated

For flow cytometry, 5 × 10^5^ HeLa cells were seeded in 6-well plates 24 h before use, intoxicated as indicated, trypsinized and resuspended in 0.5 mL of cold PBS 1x containing 2% FBS and 5 μg/mL PI to discriminate live cells. At least 20,000 cells were analyzed for each sample of PFO-treated cells. Cell were analyzed in a FACSCanto II flow cytometer (BD Biosciences, San Jose, CA, USA), and the data obtained were analyzed using FlowJo (version 9.9.3, TreeStar, Ashland, OR, USA).

### 4.9. Statistical Analysis

Statistical analyses were carried out with Prism version 6.02 (GraphPad Software, La Jolla California USA, www.graphpad.com) using one-way ANOVA with Dunnett’s post-hoc analysis to compare different means in relation to a control sample and Tukey’s post-hoc analysis for pairwise comparisons of more than two different means. Two-tailed unpaired Student’s t-test was used for comparison of the means between the two samples. Two-way ANOVA with Tukey’s post-hoc analysis was used to compare the means of samples within each condition, and Šídák’s post-hoc analysis to compare each sample mean with the other sample mean in the same condition.

## Figures and Tables

**Figure 1 toxins-11-00419-f001:**
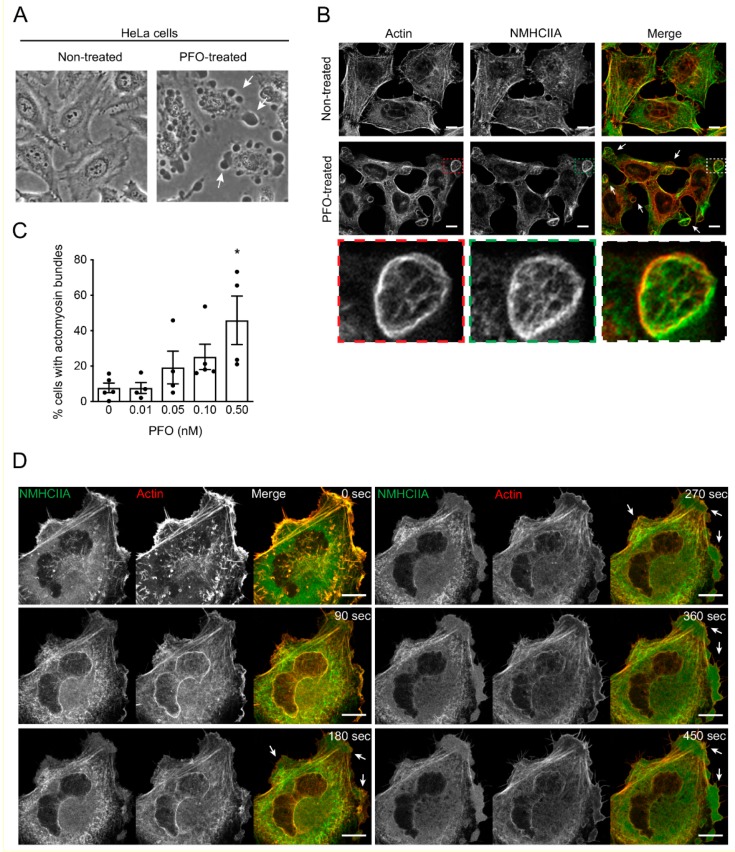
Perfringolysin O (PFO) triggers the assembly of cortical actomyosin structures in HeLa cells. (**A**) Phase-contrast microscopy images of HeLa cells left non-treated or incubated with 0.1 nM PFO for 20 min (PFO-treated). Arrows show dramatic plasma membrane (PM) blebbing induced by PFO. (**B**) Confocal microscopy images of HeLa cells left non-treated or intoxicated with PFO (0.1 nM, 20 min), stained for actin (phalloidin, red in merged images), and immunolabeled for the heavy chain of myosin IIA (NMHCIIA) (green in merged images). A cortical actomyosin structure is highlighted in insets and showed in high magnification. Scale bar: 10 μm. (**C**) Quantification of the % of cells displaying actomyosin cortical structures upon incubation with growing concentrations of PFO for 20 min. Each dot corresponds to a single independent experiment. Values are the mean ± SEM (*n* ≥ 4), p-values were calculated using one-way ANOVA with Dunnett’s post-hoc analysis (multiple comparisons to control 0 nM): * *p* < 0.5. (**D**) Sequential frames of time-lapse confocal microscopy of PFO-intoxicated HeLa cells co-expressing GFP-NMHCIIA and tdTomato-F-Tractin. PFO was added to the culture medium 20 s before t0. Arrows indicate cortical NMHCIIA bundles at PM blebbing sites.

**Figure 2 toxins-11-00419-f002:**
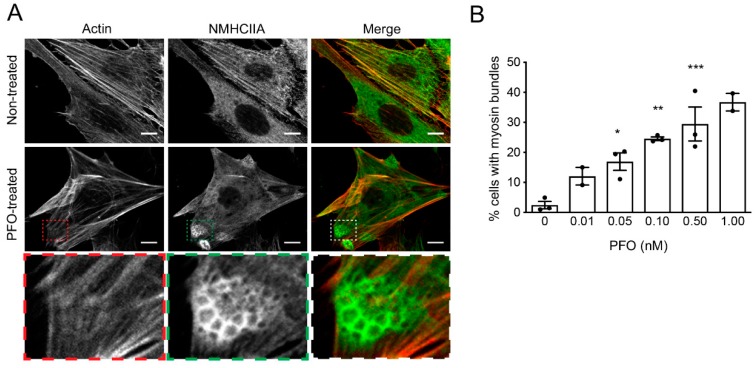
PFO intoxication of the human umbilical vein endothelial cells (HUVECs) induces the formation of cortical NMHCIIA-enriched structures. (**A**) Confocal microscopy images of HUVECs left non-treated or intoxicated with PFO (0.1 nM, 20 min), stained for actin (phalloidin, red in merged images), and immunolabeled for NMHCIIA (green in merged images). Actomyosin structures highlighted in insets are shown in high magnification images. Scale bar: 10 μm. (**B**) Quantification of the % of HUVECs displaying actomyosin structures upon incubation with increasing concentrations of PFO for 20 min. Each dot corresponds to a single independent experiment. Values are the mean ± SEM (*n* ≥ 2), *p*-values (for conditions with *n* = 3) were calculated using one-way ANOVA with Dunnett’s post-hoc analysis (0 nM as control): * *p* < 0.5; ** *p* < 0.01, and *** *p* < 0.001.

**Figure 3 toxins-11-00419-f003:**
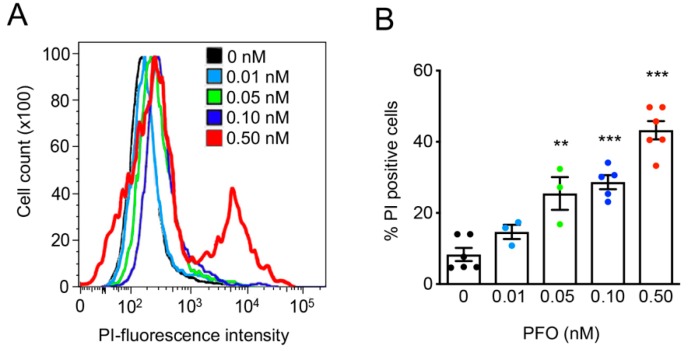
Plasma membrane permeability increases in response to increasing concentrations of PFO. HeLa cells were left untreated (0 nM) or intoxicated with increasing concentrations of PFO. Permeability of the plasma membrane was quantified by flow cytometry following propidium iodide (PI) incorporation. (**A**) Flow cytometry plots for a representative experiment are shown. (**B**) Graph displays the % of PI-positive cells obtained from flow cytometry plots, for the different conditions. Each dot corresponds to a single independent experiment. Values are the mean ± SEM (*n* ≥ 3) and *p*-values were calculated using one-way ANOVA with Dunnett’s post-hoc analysis (0 nM as control): ** *p* < 0.01, *** *p* < 0.001.

**Figure 4 toxins-11-00419-f004:**
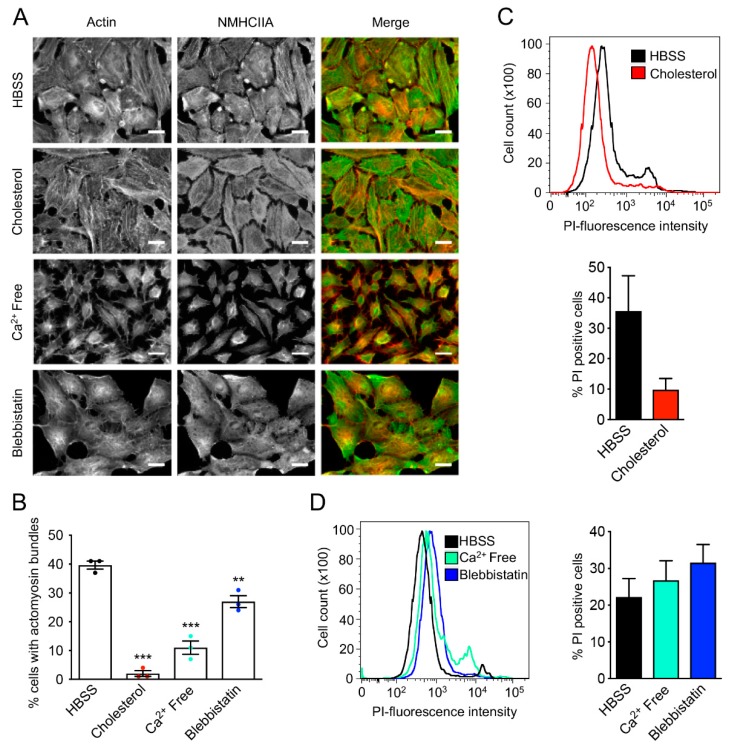
Actomyosin structures assembly in response to the PFO-induced PM damage requires pore formation, calcium influx, and myosin II activity. (**A**) Confocal microscopy images of HeLa cells incubated with PFO (0.1 nM, 20 min) in different conditions: in HBSS (control), with PFO pre-incubated with cholesterol, in Ca^2+^-free medium or using cells pre-treated with blebbistatin. Cells were stained for actin (phalloidin, red in merged images) and immunolabeled for NMHCIIA (green in merged images). Scale bar: 20 μm. (**B**) Quantification of the % of PFO-intoxicated cells displaying actomyosin structures, under the different conditions. Values are the mean ± SEM (*n* = 3, each dot shows an independent experiment), *p*-values were calculated using one-way ANOVA with Dunnett’s post-hoc analysis: ** *p* < 0.01 and *** *p* < 0.001. (**C**,**D**) Permeability of the PM quantified by flow cytometry following PI incorporation. The % of PI-positive cells for the different conditions is shown and compared to the control condition (HBSS). Flow cytometry plots for a representative experiment are shown. Graphs show quantifications of PI-positive cells obtained from flow cytometry plots. Values are the mean ± SEM (*n* ≥ 3,) and *p*-values were calculated using two-tailed unpaired Student’s *t*-test (**C**) and one-way ANOVA with Dunnett’s post-hoc analysis (**D**).

**Figure 5 toxins-11-00419-f005:**
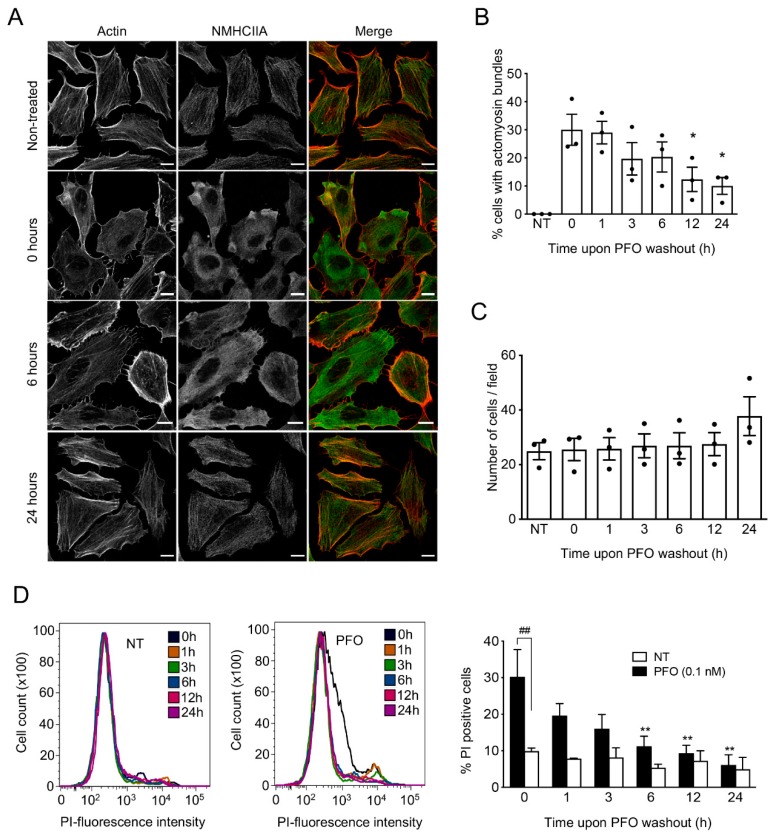
Actomyosin structures disassemble along with the recovery of PM integrity, upon PFO washout. (**A**) Confocal microscopy images of HeLa cells left non-treated or intoxicated with PFO (0.1 nM, 10 min) and allowed to recover for several hours (0, 6 and 24 h) after PFO washout. Actin was stained with phalloidin (red in merged images) and NMHCIIA was immunolabeled (green in merged images). Scale bar: 10 μm. (**B**) Quantification of the % of cells displaying actomyosin cortical structures in non-treated cells (NT) and in intoxicated cells over time after PFO washout. Values are the mean ± SEM (*n* = 3), *p*-values were calculated using one-way ANOVA with Dunnett’s post-hoc analysis (NT as control): * *p* < 0.5. (**C**) Quantification of the total number of cells in non-treated (NT) and intoxicated conditions throughout the time of recovery. (**D**) Permeability of the plasma membrane quantified by flow cytometry following PI incorporation overtime of recovery. Flow cytometry plots for a representative experiment are shown. Graph shows quantifications of PI-positive cells obtained from flow cytometry plots of non-intoxicated (NT) or intoxicated (PFO 0.1 nM) samples. Values are the mean ± SEM (*n* ≥ 3) and *p*-values for data corresponding to PFO-treated cells were calculated using two-way ANOVA with Tukey’s post-hoc analysis: ** *p* < 0.01. Comparison between NT and PFO-treated samples were calculated using two-way ANOVA with Šídák’s post-hoc analysis: ^##^
*p* < 0.05.

**Figure 6 toxins-11-00419-f006:**
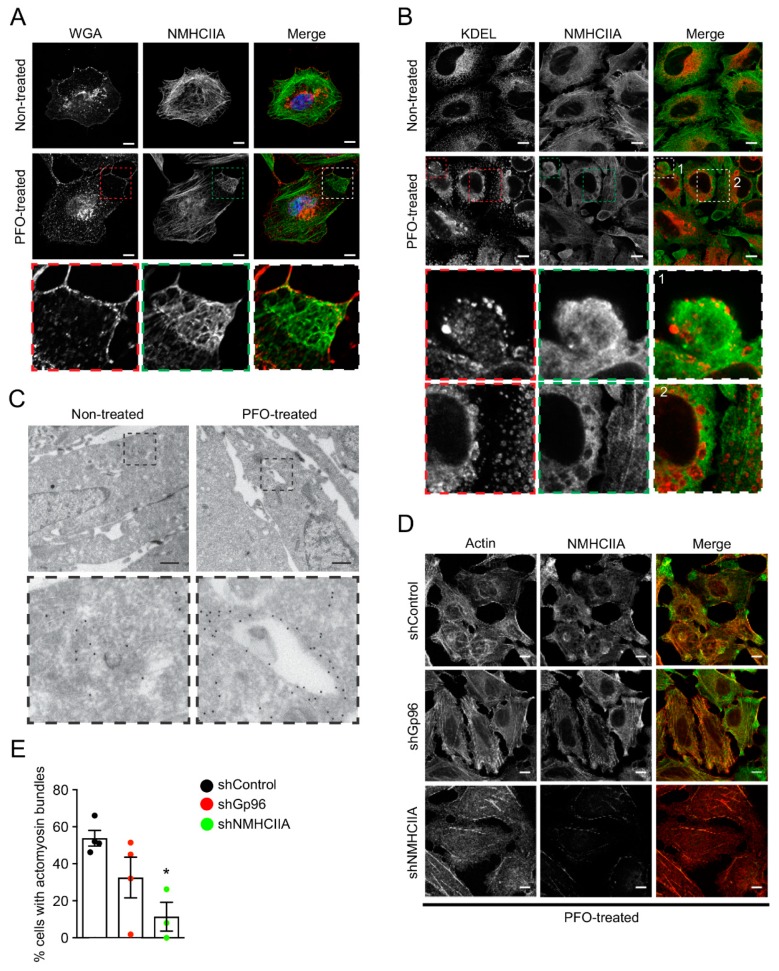
PFO induces the disruption of the ER and the accumulation of ER-vacuoles within cortical actomyosin structures. (**A**,**B**) Confocal microscopy images of HeLa cells left non-treated or intoxicated with PFO (0.1 nM, 20 min). (**A**) Plasma membrane was stained using FITC-conjugated wheat germ agglutinin (WGA) (red in merged images, false color) and NMHCIIA was immunolabeled (green in merged images, false color) (Scale bar: 15 μm). A cortical actomyosin structure is highlighted in insets and showed in high magnification. (**B**) Fixed cells were immunolabeled to detect the ER (anti-KDEL, red in merged images) and NMHCIIA (green in merged images) (Scale bar: 10 μm). A cortical actomyosin structure is highlighted in inset 1 and ER-vacuoles are shown in inset 2. Both insets are showed in high magnification images. (**C**) Transmission electron microscopy images from Lys-Asp-Glu-Leu (KDEL)-GFP overexpressing HeLa cells left non-treated or intoxicated with PFO (0.2 nM, 20 min). The ER was detected by gold-immunolabeling using anti-GFP antibody. Scale bar: 0.5 μm. Insets show high magnification images (Scale bar: 0.2 μm) of specific regions, including an ER-enriched vacuole in PFO-intoxicated cells. (**D**) Confocal microscopy images of shControl, shGp96 or shNMHCIIA HeLa cells intoxicated with PFO (0.1 nM, 20 min), stained for actin (phalloidin, red in merged images) and immunolabeled for NMHCIIA (green in merged images). Scale bar: 10 μm. (**E**) Quantification of the % of shControl, shGp96 and shNMHCIIA HeLa cells displaying actomyosin cortical structures induced by PFO intoxication (0.1 nM, 20 min). Values are the mean ± SEM (*n* ≥ 3), *p*-values were calculated using one-way ANOVA with Dunnett’s post-hoc analysis: * *p* < 0.5.

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
