# Peer review of "Perfringolysin O-Induced Plasma Membrane Pores Trigger Actomyosin Remodeling and Endoplasmic Reticulum Redistribution"

_toxins, 2019, doi:10.3390/toxins11070419_

Round 1
Reviewer 1 Report
The authors of current manuscript describe the host cytoskeletal response to perfringolysin O attack. They demonstrate that PFO at low concentration induces reorganization of the actomyosin cytoskeleton leading to the assembly of actomyosin-enriched structures. The results are described in detail and may be important for clearly understanding the mechanisms that are responsible for Clostridium perfringens infection. This manuscript is written very well and should be accepted for publication after minor modifications. Below are my detailed suggestions to improve the quality of the manuscript:
1. Why the toxin names are written with a capital letter? (lanes 29, 32,33)
2. Lane 126: assessed instead of assess; lane 190: explored instead of explore?
3.Fig. 3A. Is it possible to demonstrate the flow cytometry plot in color? This figure will be more readable if the authors color it (as demonstrated, for example, in the Fig.4D)
4.The authors demonstrate that actomyosin structures dissasemble following PFO washout, indicating that the assembly of actomyosin structures is a reversible process. We know that the level of dissociation of PFO is very low. However, can the authors demonstrate that washing the cells does not change the amount of PFO bound to the plasma membrane?
Reviewer 2 Report
This MS describes a Clostridium perfringens toxin (Perfringolysin O; PFO) has the ability to induce plasma membrane pores. These PFO induced, uncontrolled pores will further trigger actomyosin remodeling and endoplasmic reticulum redistribution. By using immunofluorescence, transmission electron microscopy and flow cytometry, this MS clearly establishes a connection between PFO and host cytoskeletal responses. This MS is well written, and this reviewer didn’t find any major problem. Some minor comments are as follows: (1) In immunofluorescence images, green and red images are shown as grey. The authors may use the grey color to improve the contrast of corresponding images. However, in the figure legends of MS, the terms red and green are still used to describe individual proteins [for instance, HeLa cells left non-treated or intoxicated with PFO (0.1 nM, 20 min), stained for actin (phalloidin, red) and immunolabeled for NMHCIIA (green)]. This may confuse the readers. Therefore, this reviewer suggests that immunofluorescence images could be presented as their original colors. 2. (Line 372) Despite adding a reference (i.e. Ref 42), the protocol of PFO purification should be briefly described.Author Response
Please see the attachment.
